# Underlying Topography Estimation over Forest Using Maximum a Posteriori Inversion with Spaceborne Polarimetric SAR Interferometry

**Xiaoshuai Li** [1,2,3], **Xiaolei Lv** [1,2,3,*] and **Zenghui Huang** [1,2,3]

1    Key Laboratory of Technology in Geo-Spatial Information Processing and Application System, Aerospace Information Research Institute, Chinese Academy of Sciences, Beijing 100190, China; lixiaoshuai21@mails.ucas.ac.cn (X.L.); huangzenghui20@mails.ucas.ac.cn (Z.H.)
2    Aerospace Information Research Institute, Chinese Academy of Sciences, Beijing 100094, China
3    School of Electronic, Electrical and Communication Engineering, University of Chinese Academy of Sciences, Beijing 100049, China
*    Correspondence: academism2017@sina.com

**Abstract:** This paper presents a method for extracting the digital elevation model (DEM) of forested areas from polarimetric interferometric synthetic aperture radar (PolInSAR) data. The method models the ground phase as a Von Mises distribution, with a mean of the topographic phase computed from an external DEM. By combining the prior distribution of the ground phase with the complex Wishart distribution of the observation covariance matrix, we derive the maximum a posterior (MAP) inversion method based on the RVoG model and analyze its Cramer–Rao Lower Bound (CRLB). Furthermore, considering the characteristics of the objective function, this paper introduces a Four-Step Optimization (FSO) method based on gradient optimization, which solves the inefficiency problem caused by exhaustive search in solving ground phase using the MAP method. The method is validated using spaceborne L-band repeat-pass SAOCOM data from a test forest area. The test results for FSO indicate that it is approximately 5.6 times faster than traditional methods without compromising accuracy. Simultaneously, the experimental results demonstrate that the method effectively solves the problem of elevation jumps in DEM inversion when modeling the ground phase with the Gaussian distribution. ICESAT-2 data are used to evaluate the accuracy of the inverted DEM, revealing that our method improves the root mean square error (RMSE) by about 23.6% compared to the traditional methods.

**Keywords:** digital elevation model (DEM); PolInSAR; maximum a posteriori estimation; Von Mises distribution; RMSprop

## 1. Introduction

According to statistics, the world has a total forest aera of 4.06 billion hectares (ha), which is 31% of the total land area [1]. The acquisition of forest understory digital elevation model (DEM) has been a challenging problem. Accurate DEM data can assist decision makers in effectively planning forest land utilization, afforestation, and logging activities, thereby achieving the conservation and optimal utilization of forest resources [2]. Meanwhile, the forest understory DEM is crucial for identifying landslides and debris flows [3]. Through the analysis of accurate DEM data, we are able to pinpoint the location and path of potential landslides and debris flows, aiding in the implementation of effective disaster prevention measures.

Light Detection and Ranging (LiDAR) and Polarimetric Interferometric Synthetic Aperture Radar (PolInSAR) are the primary means in remote sensing for obtaining DEM under the forest. LiDAR can penetrate the forest canopy and has good accuracy in estimating DEM [4–6]. However, it is limited by the atmosphere, mist and clouds. Due to high cost and

spatial discontinuity, it is difficult to achieve global and long-term coverage [7]. PolInSAR combines the sensitivity of InSAR to the spatial and height distribution of scatterers with the sensitivity of PolSAR to the shape and orientation of scatterers. It has advantages including all-weather and all-season functionality, as well as high temporal and spatial resolution, making it widely used in obtaining DEM [8–14].

To obtain the required physical parameters, it is necessary to establish a model that correlates the PolInSAR data with these parameters. In recent years, more and more models have emerged for retrieving vegetation parameters from PolInSAR data. Yamada [15] assumed that there were two uncorrelated scattering centers in the forest area, located at the top of the canopy and on the ground, respectively. This assumption transformed the problem into the direction-of-arrival (DOA) estimation. The interference phase of the ground and canopy can be obtained by ESPRIT (Estimation of Signal Parameters via Rotational Invariance Technique) [16–19]. However, because of the strong volume scattering component, the detection accuracy of this technique becomes worse for dense forest regions [20,21]. Based on the theory of electromagnetic scattering, Treuhaft et al. proposed the Random Volume over Ground (RVoG) model [22,23]. The model simplifies the forest region into a layer of randomly oriented uniform particles on the ground and derives the expression for complex interference coherence. It is a function of four physical parameters: the forest height that determines the thickness of the volume layer; the mean extinction coefficient that represents the attenuation of electromagnetic waves through the canopy; the ground phase related to the underlying topography; and the ground-to-volume amplitude ratio (GVR) that varies with polarization channels.

Solving the RVoG model parameters is a difficult problem, especially for the ground phase, which is the basis for solving the other parameters and an important parameter for inverting the DEM. In the RVOG model, the complex interference coherence can be expressed as a line in the complex plane. This line intersects the unit circle with two points, one of which represents the ground phase and the other has no practical physical meaning. This phenomenon is known as the double-candidate effect of ground phase. Based on this geometric property, Cloude et al. [24] propose a three-stage inversion (TSI) method. This method divides the inversion process into three stages: (1) least squares line fit; (2) vegetation bias removal; (3) height and extinction estimation. From the perspective of estimation theory, Tabb et al. [25,26] proposed using a maximum likelihood estimation (MLE) to retrieve forest parameters. This approach begins with the observation that the covariance matrix of the PolInSAR data follows a complex Wishart distribution. By maximizing the log-likelihood function, an objective function related solely to the ground phase is obtained. MLE provides the ability to completely separate volume scattering from surface scattering. However, the objective function regarding ground phase in MLE usually presents two indistinguishable peaks, corresponding to the two intersections of the coherence line and the unit circle in the TSI, making this method also suffers from the double-candidate effect.

Due to the double-candidate effect, the solution of the ground phase becomes complex. The three-stage method often uses some assumptions based on physical scattering to solve this problem, such as selecting the intersection point that is far away from the complex coherence under HV polarization as the ground phase [24]. However, these methods do not fully meet the assumptions in practice. Therefore, the accuracy of parameter inversion is limited. Some researchers propose solving the ground phase using the relationship between the vertical wavenumber (kz) and two candidate phases [27], which performs well with airborne data but is less effective with certain spaceborne data. For MLE, Huang et al. [28] proposed the maximum a posteriori inversion method for the RVoG model by modeling the ground phase as a Gaussian distribution (MAPG) with the mean of the topographic phase computed from the external DEM. This method effectively suppressed the double-candidate effect. However, the assumption of Gaussian distribution cannot satisfy the circular data, which may lead to incorrect results when the topographic phase appears near the phase jump point.

To make the MAP method more robust, this paper proposes a maximum a posteriori with the von Mises distribution as the prior (MAPV) method for solving the RVoG model. The method models the ground phase as a Von Mises distribution [29–31], which effectively addresses the problems caused by modeling the ground phase with the Gaussian distribution, making the estimation of the ground phase more continuous and accurate. Additionally, since the solution method for MAP requires exhaustive searching of the phase, this will significantly decrease computational efficiency. This paper proposes a four-step optimization (FSO) method for solving ground phase based on the characteristics of the MAPV objective function, employing a gradient-based optimization approach [32,33]. This method markedly improves solving efficiency without sacrificing accuracy.

This paper is organized as follows. Section 2 provides a detailed description of the experimental materials. Section 3 introduces the RVoG model and provides a brief overview of the three-stage inversion process. Section 4 presents the proposed MAPV method, and analyzes and simulates its CRLB. Furthermore, a four-step optimization method is proposed. Results and discussions are presented in Section 5. Finally, conclusions are given in Section 6.

## 2. Materials

The test area is located in the central region of Sardinia (39°48′N and 8°45′E), which is in the southwest of Italy. The area is heavily forested, with the predominant vegetation being holm oak woods. The test area consists of mountains and plains, with ground elevations ranging from 0 to 760 m above mean sea level. Figure 1a depicts the Google Earth optical image of the area projected onto the SAR coordinate system, with the dark green area indicating vegetated regions.

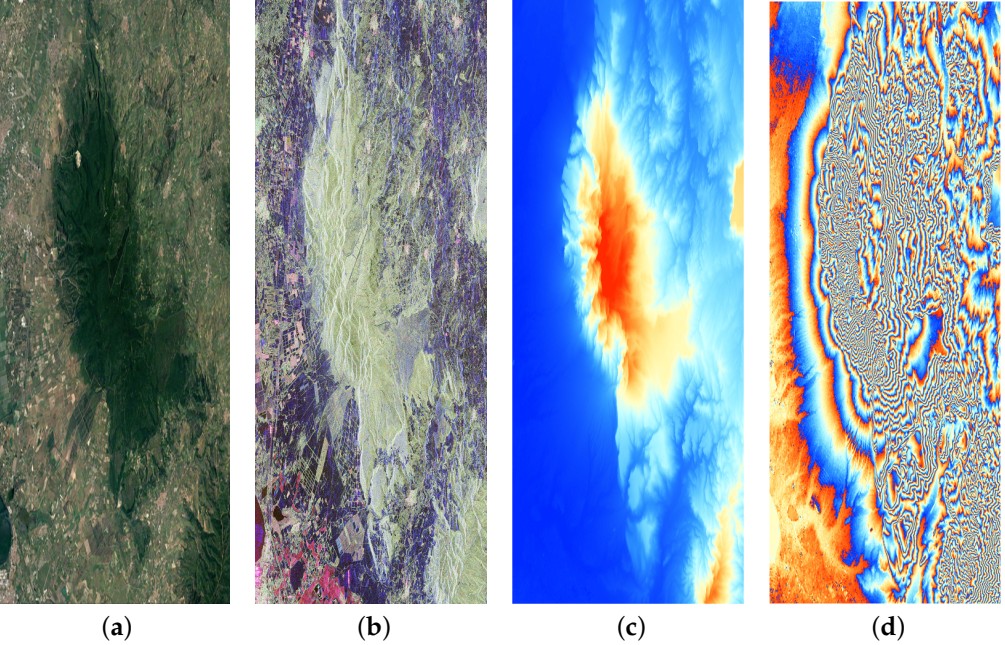

|  (a)  |  (b)  |  (c)  |  (d)  |

**Figure 1.** (**a**) Optical image of the test area in Google Earth. (**b**) L-band SAR image in the Pauli basis (R: HH - VV; G: HV; B: HH + VV) (**c**) Alos DEM projected to SAR coordinates. (**d**) The topographic phase estimated by Alos DEM.

Paired quad-polarized data used for interferometry were acquired by the SAOCOM-1A and SAOCOM-1B satellites, respectively. The SAOCOM (Satélite Argentino de Observación COn Microondas) satellite series is developed by the Argentine Space Agency and comprises two satellites: SAOCOM 1A and SAOCOM 1B. This mission provides L-band (approximately 1.275 GHz) full polarimetric data with spatial resolutions ranging from 10 to 100 m in both real-time and stored modes, and an incidence angle between 20 to 50°.

Figure 1b displays a Pauli-based PolSAR image of the test area, with the RGB channels representing HH-VV, HV, and HH+VV, respectively.

Height checkpoints are from Ice, Cloud and land Elevation Satellite-2 (ICESat-2) data [34]. This satellite carries the Advanced Topographic Laser Altimeter System (ATLAS), which introduced single photon detection technology for the first time in Earth's elevation measurement, significantly improving the data acquisition rate for terrain detection. This article employs the land and vegetation height data (ATL08) from ICESat-2, with a vertical uncertainty of 0.2 m for flat terrain and 2.0 m for mountainous terrain [35].

The external DEMs selected for the experiment are the ALOS Global Digital Surface Model "ALOS World 3D - 30 m" (AW3D30) and the Shuttle Radar Topography Mission 30 m (SRTM30), both datasets having an approximate resolution of 30 m. Figure 1c,d show the Alos DEM projected to the SAR coordinate system and the topographic phase, respectively.

Figure 2 shows the flowchart of the proposed DEM inversion method. The entire inversion framework consists of (a) ground phase estimation based on MAPV and (b) DEM generation.

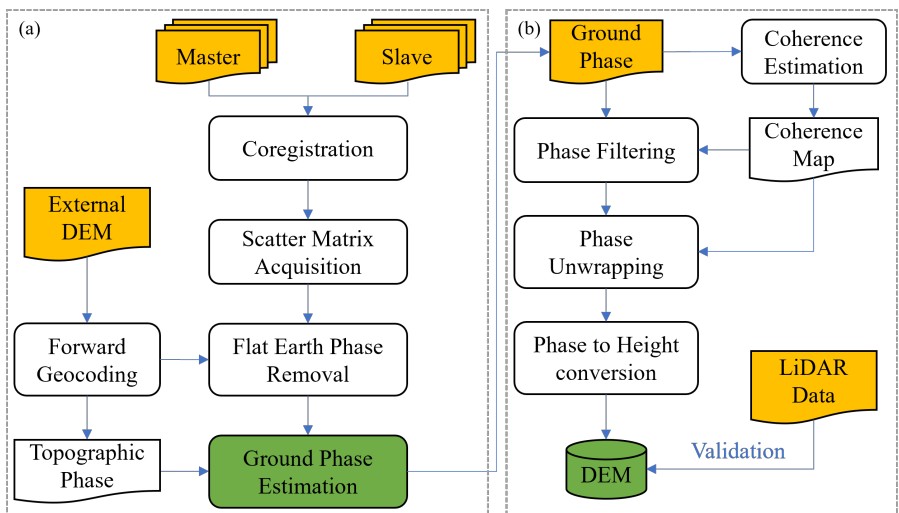

**Figure 2.** The DEM inversion flowchart for spaceborne PolInSAR. (**a**) Ground phase estimation based on MAPV. (**b**) DEM generation.

## 3. Scattering Model

### 3.1. PolInSAR Data Description

A single-baseline PolInSAR system acquires the complex scattering matrix of each resolution element via dual-polarized antennas $(h, v)$ from two slightly different look angles. According to the reciprocity $(S_{hv} = S_{vh})$, the scattering matrix can be expressed in the form of Pauli vector [36], defined as

$$\vec{k}_i = \frac{1}{\sqrt{2}} \begin{bmatrix} S_{hh_i} + S_{vv_i} & S_{hh_i} - S_{vv_i} & 2S_{hv_i} \end{bmatrix}^T, i = 1, 2 \tag{1}$$

where the $S_{pq}$ represents the scattering coefficients, $p, q \in \{h, v\}$, and the superscript $(\cdot)^T$ denotes the vector transpose operation.

The $6 \times 6$ covariance matrix can be defined as

$$\hat{R} = \left\langle \begin{bmatrix} \vec{k}_1 \\ \vec{k}_2 \end{bmatrix} \begin{bmatrix} \vec{k}_1^H & \vec{k}_2^H \end{bmatrix} \right\rangle = \begin{bmatrix} \hat{T}_1 & \hat{\Omega} \\ \hat{\Omega}^H & \hat{T}_2 \end{bmatrix} \tag{2}$$

where $(\cdot)^H$ is the conjugate transpose, and $\langle \cdot \rangle$ is the average operation in data processing. $\hat{T}_1$ and $\hat{T}_2$ contain the polarimetric information of the two images, respectively, called the polarimetric coherence matrix, while $\hat{\Omega}$ contains the polarimetric and interferometric information of the two images, called the polarimetric inteferometry matrix.

For any given non-zero scattering mechanism $\boldsymbol{\omega}$, the complex interferometric coherence can be expressed as

$$\gamma(\boldsymbol{\omega}) = \frac{\boldsymbol{\omega}^H \hat{\boldsymbol{\Omega}} \boldsymbol{\omega}}{\boldsymbol{\omega}^H \hat{\boldsymbol{T}} \boldsymbol{\omega}} \tag{3}$$

where $\hat{\boldsymbol{T}} = (\hat{\boldsymbol{T_1}} + \hat{\boldsymbol{T_2}})/2$.

*3.2. The Random Volume over Ground (RVoG) Model*

Without considering other sources of decorrelation, such as system and temporal factors, and focusing solely on the decorrelation effect induced by scatterers, the complex coherence (3) can be expressed as the integral ratio form of the vertical distribution function $F(z)$ of the scatterers [37]

$$\gamma(\boldsymbol{\omega}) = e^{jk_z z_0} \frac{\int_0^{h_v} F(z)e^{jk_z z}dz}{\int_0^{h_v} F(z)dz} \tag{4}$$

where $z_0$ is the vertical position of the ground surface, $h_v$ reperesents the forest height, and $k_z$ denotes the wavenumber.

The RVoG model assumes that the forested area is composed of randomly oriented uniform particles, with an exponential vertical distribution function, and that the ground layer is impenetrable with a delta function for its vertical distribution, defined as

$$F(z) = m_v(\boldsymbol{\omega})e^{\frac{2\sigma z}{\cos \theta_0}} + m_g(\boldsymbol{\omega})\delta(z) \tag{5}$$

where $m_v$ and $m_g$ are the volume and the ground scattering amplitudes, respectively. $\theta_0$ is the incidence angle, and $\sigma$ represents the mean extinction coefficient. By inserting (5) into (4), we can obtain

$$\gamma(\boldsymbol{\omega}) = e^{j\phi_0} \frac{\gamma_v + \mu(\boldsymbol{\omega})}{1 + \mu(\boldsymbol{\omega})} = e^{j\phi_0}(\gamma_v + \frac{\mu(\boldsymbol{\omega})}{1 + \mu(\boldsymbol{\omega})}(1 - \gamma_v)) \tag{6}$$

where $\gamma_v$ denotes the volume coherence, $\phi_0$ denotes the ground phase, and $\mu(\boldsymbol{\omega})$ represents the GVR.

Notice that (6) represents the complex interference coherence as a straight line on the complex plane. The three-stage inversion method mentioned above is based on this geometric property. This method fits the complex coherence into a straight line that intersects the unit circle at two points. The ground phase is then selected from these two points based on the feature that the direct ground scattering signal is weak in the L-band HV channel [24].

**4. Maximum a Posteriori Estimation of the Ground Phase**

Given the observed value $\hat{\boldsymbol{R}}$, the maximum a posteriori (MAP) estimate of the parameter $\phi$ can be expressed as

$$\begin{aligned} \hat{\phi}_{\text{MAP}} &= \arg\max_{\phi} \ p(\phi|\hat{\boldsymbol{R}}) \\ &\propto \arg\max_{\phi} \ p(\hat{\boldsymbol{R}}|\phi)p(\phi) \\ &\propto \arg\max_{\phi} \{\log p(\hat{\boldsymbol{R}}|\phi) + \log p(\phi)\} \end{aligned} \tag{7}$$

where $p(\hat{\boldsymbol{R}}|\phi)$ is the likelihood function, which represents the performance of the data set, and $p(\phi)$ represents a prior distribution of $\phi$.

It is known that the observed coherence matrix of PolInSAR after multilook (2) follows a complex wishart distribution

$$p(\hat{\boldsymbol{R}}; \boldsymbol{R}, N) = c(\hat{\boldsymbol{R}})|\boldsymbol{R}|^{-N} \exp(-N \cdot tr(\boldsymbol{R}^{-1}\hat{\boldsymbol{R}})) \tag{8}$$

where $N$ is the number of looks, $c(\hat{R})$ is a constant for normalization, $|\cdot|$ is the determinant operator, and $tr$ is the trace operator.

A priori knowledge about the ground phase is obtained by introducing an external DEM.

*4.1. A Prior Probability Model of the Ground Phase*

With InSAR geometry, the topographic phase can be solved by an external DEM [38]

$$\phi_{\text{topo}} = -\frac{4\pi B_{\perp}}{\lambda R \sin\theta} h. \tag{9}$$

The ground phase in PolInSAR is the same as the topographic phase, both representing the contribution of the underlying ground to the interferometric phase. Considering the influence of vertical accuracy and resolution, there is an error between the topographic phase calculated from an external DEM and the actual ground phase. The ground phase can be represented as:

$$\phi = \phi_{\text{topo}} + \phi_e \tag{10}$$

where $\phi_e$ denotes the error component. Some researchers modeled the $\phi_e$ as a Gaussian distribution with zero mean and variance dependent on the external DEM. Therefore, the ground phase follows a Gaussian distribution with a mean of the $\phi_{\text{topo}}$. The maximum a posteriori estimation method was used to solve the RVoG model parameters. This method effectively suppressed the double-candidate effect, making the two local maxima of the objective function clearly distinguishable [28].

However, since the estimated ground phase is the principal value of the phase within the $[-\pi, \pi)$ interval after the periodic wrapping, $\pi$ and $-\pi$ differ numerically by $2\pi$ but represent the same point in the complex plane. This point is called the phase jump point. The assumption of the Gaussian distribution does not take into account the characteristic of phase wrapping, so it is discontinuous near the phase jump point, which can lead to errors in phase estimation.

Figure 3a,c show the Gaussian distribution curves in the Cartesian coordinate system and on the unit circle of the complex plane, respectively, for one phase period. In the complex plan, it is evident that among the two candidates ($c_1$ and $c_2$), $c_2$ is closer to $\phi_{\text{topo}}$, so it should have a higher probability of being the ground phase. However, in Figure 3a, the probability of $c_1$ being the ground phase is greater than that of $c_2$, which is obviously incorrect.

The modeling of the ground phase must consider the characteristic of phase wrapping while retaining the selectivity of the Gaussian distribution. At present, probability models have not found much application to circular data. Common ones include: Uniform, Cardioid, Wrapped Normal, Wrapped Cauchy, and Von Mises distributions [29,39]. This paper employs the Von Mises distribution to model ground phase, offering significant advantages in this context. Firstly, its circular nature enables better adaptation to periodic data, akin to Cardioid and Wrapped Normal distributions, but more flexible than Wrapped Cauchy distribution. Secondly, the Von Mises distribution possesses parametric flexibility, enabling adjustment of its parameters to accommodate different degrees of data concentration. Furthermore, the mathematical form of the Von Mises distribution is simple and easy to understand, facilitating straightforward computation and interpretation, while also possessing robust statistical properties such as mean, variance, etc., making it more manageable and analyzable in practical applications [30,40].

The probability density function of the Von Mises distribution is

$$p(\phi; \phi_{\text{topo}}, \kappa) = \frac{e^{\kappa \cos(\phi - \phi_{\text{topo}})}}{2\pi I_0(\kappa)} \tag{11}$$

where $\kappa$ represents the degree of concentration (similar to the inverse of variance), $I_0(\kappa)$ is the modified Bessel function of the first kind of order 0, which is used for normalization.

As can be seen in Figure 3b,d, the Von Mises distribution effectively solves the problem of discontinuity at phase jump point in the Gauss distribution, making the ground phase estimation more robust.

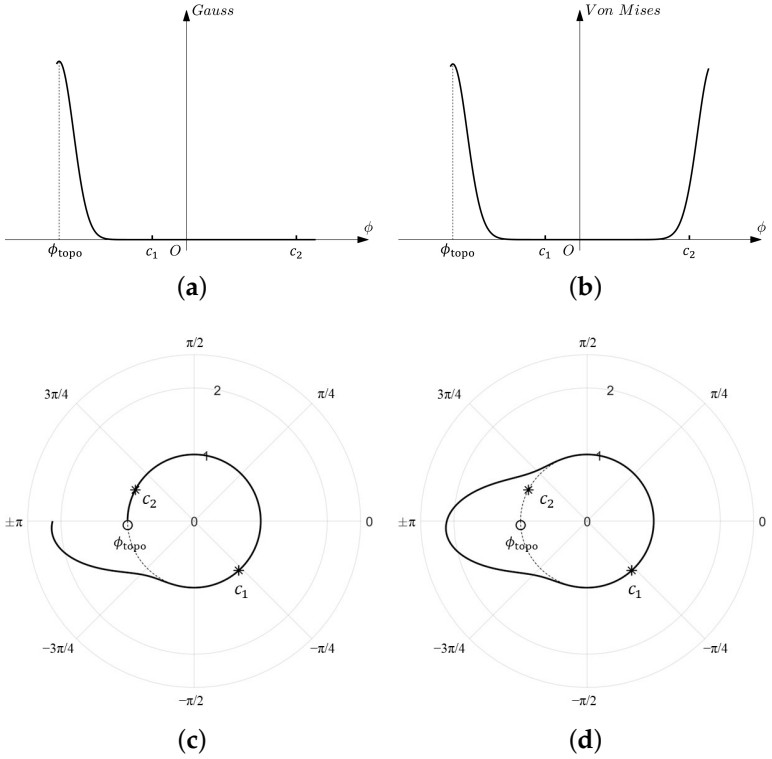

**Figure 3.** The difference between Gauss distribution and Von Mises distribution. Up: In the Cartesian coordinate system. (**a**) Gauss distribution. (**b**) Von Mises distribution. Down: In polar coordinate. (**c**) Gauss distribution. (**d**) Von Mises distribution. The hollow black point represents the position of the topographic phase on the unit circle of the complex plane, and the asterisks correspond to the phases at the two peaks of the MLE.

*4.2. MAP with Von Mises Distribution as Prior*

By changing $m_v(\boldsymbol{\omega})$ and $m_g(\boldsymbol{\omega})$ in (5) to matrix form

$$
\begin{aligned}
m_v(\boldsymbol{\omega}) &= \boldsymbol{\omega}^H \boldsymbol{T}_v \boldsymbol{\omega} \\
m_g(\boldsymbol{\omega}) &= \boldsymbol{\omega}^H \boldsymbol{T}_g \boldsymbol{\omega}
\end{aligned}
\tag{12}
$$

and inserting them into (4), we can obtain the matrix form of the RVoG model in the same form of (3), and

$$
\begin{aligned}
\boldsymbol{T} &= I_1 \boldsymbol{T}_v + a \boldsymbol{T}_g \\
\boldsymbol{\Omega} &= e^{j\phi}(I_2 \boldsymbol{T}_v + a \boldsymbol{T}_g)
\end{aligned}
\tag{13}
$$

where $a$, $I_1$, $I_2$ are functions of the forest height $h_v$ and extinction coefficient, $\boldsymbol{T}_v$ and $\boldsymbol{T}_g$ are the volume and ground coherence matrices, respectively. The volume only coherence can be obtained by $\gamma_v = I_2/I_1$.

It is worth noting that $\hat{\boldsymbol{R}}$, $\hat{\boldsymbol{T}}$, and $\hat{\boldsymbol{\Omega}}$ are the ensemble averages of the data and should be distinguished from the ideal mathematical expectations $\boldsymbol{R}$, $\boldsymbol{T}$, and $\boldsymbol{\Omega}$.

Combining the complex Wishart distribution (8) and the prior probability distribution (11), the objective function of the MAPV estimation of the RVoG model is given by (7)

$$
f_{\text{MAPV}} = \log p(\hat{\boldsymbol{R}}; \boldsymbol{R}, N) + \log p(\phi; \phi_{\text{topo}}, \kappa)
\tag{14}
$$

where the first term is the objective function of the MLE. Maximising it yields [25]

$$\log p(\hat{\pmb{R}}; \pmb{R}, N) = N \cdot (3 \log(1 - \cos \theta) - \log |\pmb{A}_{\theta+\phi}| - \log |\pmb{A}_{\phi}|) \tag{15}$$

where

$$\pmb{A}_{\pmb{\alpha}} = \hat{\pmb{T}} - \frac{1}{2}(e^{-j\alpha}\hat{\pmb{\Omega}} + e^{j\alpha}\hat{\pmb{\Omega}}^H) \tag{16}$$

and $\theta$ represents the angle of the intersection of the unit circle with the extension of the line segment connecting 1 and $\gamma_v$ in the complex plane, geometrically.

Substituting (11) and (15) into (14), the objective function with respect to the $\phi$ and $\theta$ is obtained as

$$\begin{aligned} f_{\text{MAPV}} &= 3 \log(1 - \cos \theta) - \log |\pmb{A}_{\theta+\phi}| - \log |\pmb{A}_{\phi}| \\ &\quad + \frac{\kappa}{N} \cos(\phi - \phi_{\text{topo}}). \end{aligned} \tag{17}$$

Differentiating the objective function with respect to $\theta$ and $\phi$, respectively, and setting them to 0 yields

$$\begin{aligned} f_{\theta} &= \frac{3 \sin \theta}{1 - \cos \theta} - \frac{d}{d\alpha} \log |\pmb{A}_{\pmb{\alpha}}| = 0 \\ f_{\phi} &= -\frac{d}{d\alpha} \log |\pmb{A}_{\pmb{\alpha}}| - \frac{|\pmb{A}_{\phi}|'}{|\pmb{A}_{\phi}|} - \frac{\kappa}{N} \sin(\phi - \phi_{\text{topo}}) = 0. \end{aligned} \tag{18}$$

Combining the two equations of (18), we can obtain the expression of $\theta$ with respect to $\phi$

$$\theta = 2 \tan^{-1} \left[ \frac{-3}{\frac{|\pmb{A}_{\phi}|'}{|\pmb{A}_{\phi}|} + \frac{\kappa}{N} \sin(\phi - \phi_{\text{topo}})} \right]. \tag{19}$$

Figure 4 shows the difference between the objective functions of MAPG and MAPV. By observing the positions of $\phi_{\text{topo}}$, $c_1$, and $c_2$ in the complex plane in Figure 3, it can be seen that the expected peak of the objective function should appear near $c_2$, which is consistent with the MAPV objective function.

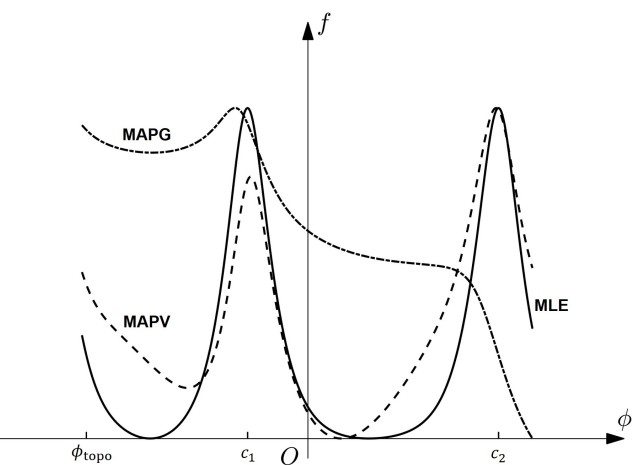

**Figure 4.** Illustration of the difference between MAPG and MAPV. The solid line, dotted line, and dashed line represent the normalized objective functions of the MLE method, MAPG method, and MAPV method, respectively.

According to (14) and (19), the ground phase is given by

$$\phi_0 = \arg\max_{\phi} \left\{ \frac{(1 - \cos \theta)^3}{|\pmb{A}_{\theta+\phi}||\pmb{A}_{\phi}|} e^{\frac{\kappa}{N} \cos(\phi - \phi_{\text{topo}})} \right\} \tag{20}$$

which is related to the underlying topography. The DEM can be obtained from part (b) of Figure 2.

### 4.3. The Cramer–Rao Lower Bound Analysis

To evaluate the validity of the method, it is necessary to analyze its Cramer–Rao Lower Bound (CRLB), which provides the minimum bound of variance if it is unbiased.

Note that the above derivation of MAPV is done under the noise-free assumption. In practice, the PolInSAR system inevitably introduces noise, the noise-affected R can be expressed as [41]

$$R_n = \begin{bmatrix} T & \Omega \\ \Omega^H & T \end{bmatrix} + \begin{bmatrix} N_1 & 0 \\ 0 & N_2 \end{bmatrix} \tag{21}$$

where $N_1$ and $N_2$ are the noise covariance matrices of the receivers at each end of the baseline. Assuming that the noise between channels is uncorrelated, $N_i$ $(i = 1, 2)$ can be expressed as

$$N_i = \frac{1}{2} \begin{bmatrix} \sigma_{hh_i}^2 + \sigma_{vv_i}^2 & \sigma_{hh_i}^2 - \sigma_{vv_i}^2 & 0 \\ \sigma_{hh_i}^2 - \sigma_{vv_i}^2 & \sigma_{hh_i}^2 + \sigma_{vv_i}^2 & 0 \\ 0 & 0 & \sigma_{hv_i}^2 + \sigma_{vh_i}^2 \end{bmatrix} \tag{22}$$

where $\sigma_{pq}^2$, $p$, $q \in \{h, v\}$ denotes the noise variance of each channel. Note that since the noise of the two receivers is uncorrelated, the noise has no effect on $\Omega$.

The CRLB can be derived from the inverse of the Fisher information. The derivation of the Fisher information is presented in Appendix A for clarity, which can be expressed as [42,43]

$$I_F = Ntr\left( R_n^{-1} \frac{\partial R_n}{\partial \phi} R_n^{-1} \frac{\partial R_n}{\partial \phi} \right) + \frac{\kappa I_1(\kappa)}{I_0(\kappa)}. \tag{23}$$

Therefore, the CRLB is

$$\text{Var}\left[\hat{\phi}\right] \geq \frac{1}{I_F} = \frac{1}{Ntr\left( R_n^{-1} \frac{\partial R_n}{\partial \phi} R_n^{-1} \frac{\partial R_n}{\partial \phi} \right) + \frac{\kappa I_1(\kappa)}{I_0(\kappa)}}. \tag{24}$$

Figure 5 shows the simulation results using the data from [44] with $N = 50$ and $\kappa = 3.65$ (approximately equivalent to 30°).

It is worth noting that the variance of the TSI method falls below the theoretical CRLB curve when the SNR is reduced below about $-14$ dB. This is due to the fact that the phase error is calculated in the sense of phase wrapping, whereas the CRLB is calculated using the local curvature of the distribution function [45]. From Figure 5, it can be seen that the inclusion of the prior effectively reduces the CRLB, and also demonstrates the superiority of the MAPV method over the TSI method for ground phase estimation.

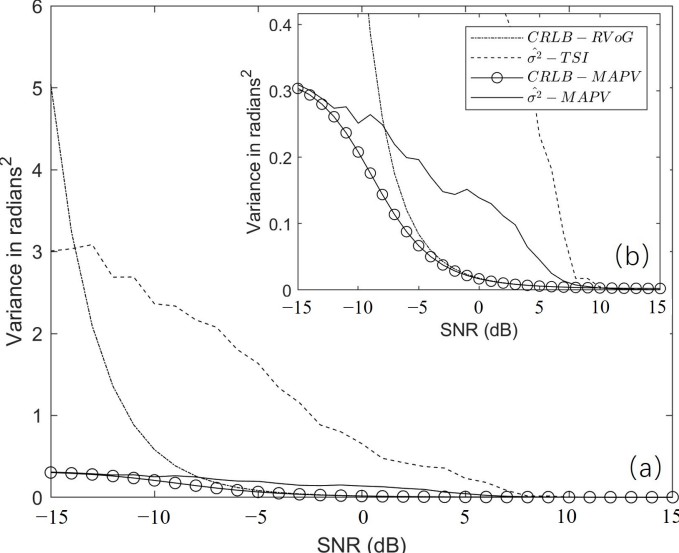

**Figure 5.** (**a**) Evolution of the CRLB with signal-to-noise ratio (SNR). (**b**) A detail extracted from (**a**).

### 4.4. Four-Step Optimized Solution for MAPV

To determine the ground phase, which corresponds to maximizing the objective function (17), the typical approach involves conducting an exhaustive search (ES) over $\phi$. Although this ensures accuracy in finding the solution, it leads to high computational complexity and low computational efficiency.

Since the objective function of MAPV with respect to $\phi$ is a continuous curve, as shown in Figure 4. Based on this property, this paper employs the gradient optimization method to enhance the efficiency of solving ground phase. The gradient of the MAPV objective function is presented in Appendix B.

Given the significant variations in the objective function for each pixel, the method requires addressing two distinct problems:

1.  Initial phase: Due to the presence of two peaks in the objective function, a single initial point is prone to fall into the local maximum. Therefore, it is necessary to choose an appropriate strategy to ensure that the results converge to the global maximum.
2.  Global learning rate: Because of the large differences between the objective functions of different pixels, it is very important to choose the global learning rate so that the method can adapt to all objective functions.

#### 4.4.1. Initial Phase

Taking into account the characteristics of the MAPV objective function depicted in Figure 6, this paper introduces a four-step solving approach.

*   Step 1: Gradient descent using $\phi_{\text{topo}}$ as initial phase. Get the $\phi_{\text{vally}}$ between the two peaks.
*   Step 2: Give the $\phi_{\text{vally}}$ a $\Delta_\phi$ in the opposite direction to $\phi_{\text{topo}}$ as a $\phi_{\text{seed}}$.
*   Step 3: Perform gradient ascent separately using this $\phi_{\text{seed}}$ and $\phi_{\text{topo}}$ to obtain the two peak values and their respective phases.
*   Step 4: Compare the magnitudes of these two peak values and select the phase corresponding to the larger peak value as the final result.

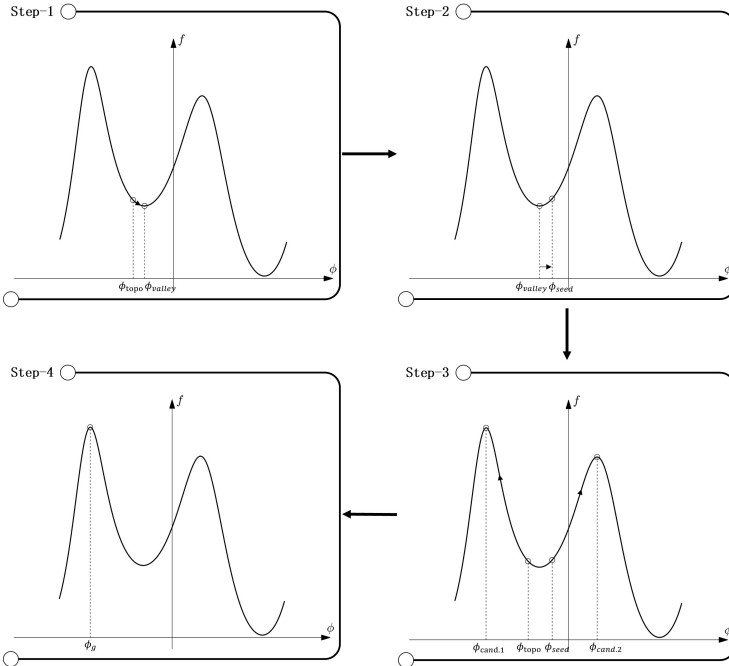

**Figure 6.** Illustration of the four-step optimization procedure for solving the ground phase.

#### 4.4.2. Global Learning Rate

In the FSO method, we employed both gradient ascent and gradient descent algorithms. For convenience, this paper converts the gradient ascent process into a gradient descent process, with the specific conversion procedure given below.

The gradient descent algorithm is based on the Root Mean Square Propagation (RM-SProp) [33]. Although RMSprop adjusts the learning rate during iterations, the global learning rate determines the speed, convergence, and eventual performance of the model during training. If the learning rate is too high, it may lead to an unstable training process and difficulties in convergence. Conversely, a low learning rate might result in slower training speed, requiring more iterations to converge to an optimal solution.

Due to significant differences in the objective functions of various pixels, selecting a uniform global learning rate poses a challenge. Therefore, this paper proposes improvements to RMSprop to reduce the model's reliance on the global learning rate to accommodate all objective functions, as shown in Algorithm 1.

When the current gradient $g$ has opposite signs with the minimum gradient $g_{min}$, or when the current function value $f(\phi)$ is greater than the minimum function value $f_{min}$, it indicates that the step size might be too large, causing oscillations. In such cases, the minimum phase is necessarily situated between $\phi$ and $\phi_{min}$, closer to the side with the lower function value. In the fourth line of Algorithm 1, a weighted computation based on function values is applied for this condition, assigning higher weight to the lower function value when updating the phase. As the updated phase is very close to the minimum phase, it is advisable to moderately decrease the learning rate for better convergence.

Algorithm 1 assumes the objective function to be positive, but neglecting certain constant terms during derivation may result in the objective function potentially becoming negative. This could result in an error on line 4 during execution. This problem can be solved by preprocessing the coherence matrix, let

$$T = E^{-\frac{1}{2}} T E^{-\frac{1}{2}}$$
$$\Omega = E^{-\frac{1}{2}} \Omega E^{-\frac{1}{2}}$$

(25)

where $E$ is the diagonal matrix formed by the eigenvalues of $T$.

---

**Algorithm 1:** Framework of improved RMSprop algorithm.

**Require:** Global learning rate $\epsilon$, Decay rate $\rho$, Initial phase $\phi$
**Initialize:** $r \leftarrow 0$, $f_{min} \leftarrow \inf$, $\phi_{min} \leftarrow 0$, $g_{min} \leftarrow 0$, $n \leftarrow 0$, $\delta \leftarrow 10^{-6}$

1 **while** *stopping criterion not met* **do**
2    Compute gradient: $g \leftarrow \nabla f(\phi)$;
3    **if** $g \cdot g_{min} < 0 \mid f(\phi) > f_{min}$ **then**
4       $\Delta_\phi \leftarrow \frac{\phi_{min}/f_{min} + \phi/f(\phi)}{1/f_{min} + 1/f(\phi)} - \phi$;
5       **if** $f_{min} > f(\phi)$ **then**
6          $f_{min}, \phi_{min}, g_{min} \leftarrow f(\phi), \phi, g$;
7       $\phi \leftarrow \phi + \Delta_\phi$;
8       $\epsilon \leftarrow \frac{\epsilon}{2 + \log(n)}$;
9    **else**
10       $f_{min}, \phi_{min}, g_{min} \leftarrow f(\phi), \phi, g$;
11       $r \leftarrow \rho \cdot r + (1 - \rho) \cdot g \cdot g$;
12       $\Delta_\phi \leftarrow \frac{\epsilon}{\sqrt{r+\delta}} \cdot g$;
13       $\phi \leftarrow \phi - \Delta_\phi$;
14    $n \leftarrow n + 1$;

---

In FSO, the gradient ascent process can be converted into a gradient descent process. In the gradient ascent task, calculate the function value $f$ and gradient $g$ at that point, then let

$$g = -g/f^2$$
$$f = 1/f.$$

(26)

Then, Algorithm 1 can be used for solving.

Although the method requires performing three iterations of gradient descent, the improvement made in RMSprop based on the characteristics of the objective function allows each gradient descent to converge to the extremum within ten iterations. Furthermore, the precision obtained by this algorithm often surpasses that achieved through exhaustive search (<1 degree).

## 5. Results

### 5.1. Evaluation Indicator

The evaluation indicators are mean error $m$, root mean square error (RMSE) $r$, and accuracy $\alpha_\sigma$

$$m = \frac{1}{N} \sum_{n=1}^{N} (\tilde{x}_n - x_n)$$

$$r = \sqrt{\frac{1}{N} \sum_{n=1}^{N} (\tilde{x}_n - x_n)^2} \qquad (27)$$

$$\alpha_\sigma = \frac{1}{N} \sum_{n=1}^{N} I_{|\tilde{x}_n - x_n| \leq \sigma}$$

where $I_{|\tilde{x}_n - x_n| \leq \sigma}$ is the idicative function, defined as

$$I_{|\tilde{x}_n - x_n| \leq \sigma} = \begin{cases} 1, & \text{if } |\tilde{x}_n - x_n| \leq \sigma \\ 0, & \text{otherwise.} \end{cases} \qquad (28)$$

### 5.2. Efficiency of FSO for MAPV

As mentioned above, the discrepancy in the objective function between pixels poses a challenge in the selection of the global learning rate when using traditional RMSprop. Algorithm 1 proposed in this paper is an improvement specifically designed to address this issue, and it is necessary to discuss its convergence here.

Figure 7 presents the convergence performance of RMSprop and Algorithm 1 under different global learning rates. As depicted in Figure 7a,d, under conditions of a relatively small global learning rate, both methods exhibit robust convergence. However, this comes at the cost of requiring more than 20 iterations. As the global learning rate increases, Algorithm 1 demonstrates a pronounced acceleration in convergence. In contrast, the RMSprop fails to converge to extrema due to oscillations.

While a larger $\epsilon$ appears to contribute to faster convergence in this experiment, it is important to note that, due to the potentially steep peaks in the MAPV objective function, excessively high global learning rates may cause the algorithm to jump out of these peaks. Therefore, it is also not advisable to set the global learning rate too high.

Table 1 presents the efficiency of the four-step optimization method for the ground phase of the test area. Here, 'ES' denotes achieving 1-degree accuracy through exhaustive search. It is evident that the proposed method is 5.6 times faster than exhaustive search while ensuring accuracy. On average, convergence is achieved in only 24.5 steps per pixel.

**Table 1.** Efficiency assessment of four-step optimization method.

| Size | Time (s) | | Iterations | | $\alpha_{1°}$ | $\alpha_{2°}$ |
|---|---|---|---|---|---|---|
| | ES | FSO | ES | FSO | | |
| $7015 \times 2673$ | 6880.0 | 1235.1 | 360 | 24.5 | 99.97% | 99.99% |

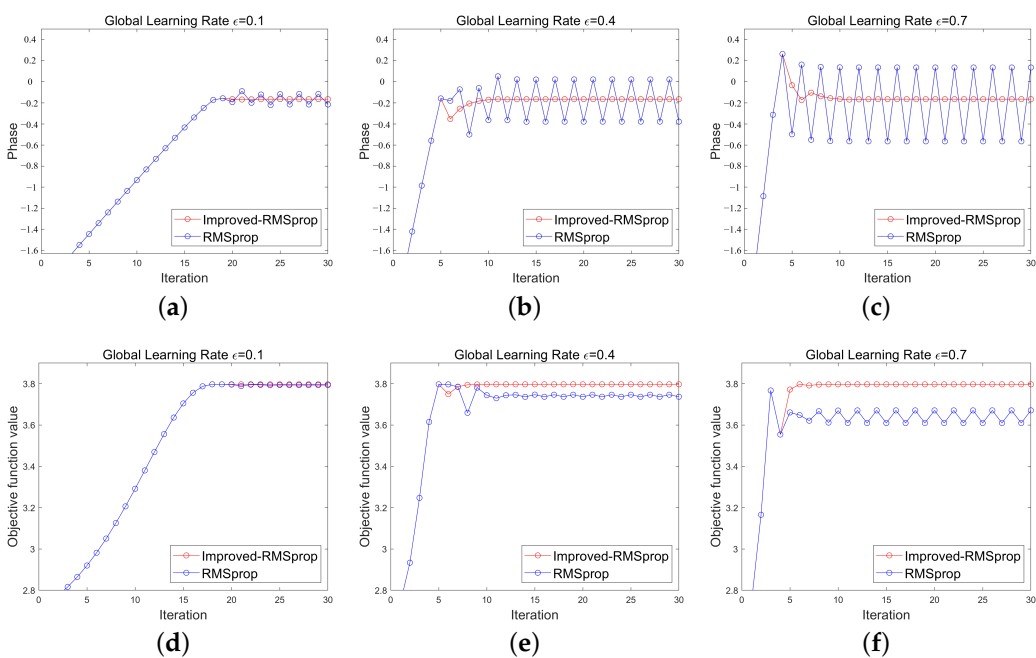

**Figure 7.** The convergence of the improved RMSprop. The top (**a**–**c**) and bottom (**d**–**f**) rows, respectively, depict the convergence of phase and objective function values when the global learning rate $\epsilon = 0.1, 0.4$, and 0.7.

### 5.3. Results of DEM Inversion in the Test Area

Given that the Gaussian distribution is not suitable for modeling the ground phase, MAPG encounters discontinuities when solving for ground phase. These discontinuities are propagated to the DEM during phase inversion, resulting in discontinuities within the DEM. This section will employ the Alos DEM as a benchmark to evaluate the extent of this discontinuity. Despite vertical accuracy differences between Alos DEM and actual values, it remains an essential reference in assessing DEM continuity.

As shown in Figure 8, the phase estimated by MAPG and MAPV show similar fringes that are basically consistent with the topographic phase computed by the external DEM. However, the comparison shows that there are significant differences between the two methods in the vicinity of the phase jump point (the blue and red boundary region in the phase diagram). In this region, the MAPG typically transitions with an intermediate value, resulting in a phase discontinuity that causes elevation jumps when inverting the DEM.

The DEM inverted by the two methods are shown in Figure 9a,b. Due to the large elevation range in the test area, it is not easy to observe differences over an entire image. Therefore, we select two areas for comparison, corresponding to the two areas in Figure 8. Different scales are set for the two areas. As shown in Figure 9, the MAPG-inverted DEM shows a strong discontinuity in some regions that are highly correlated with the vicinity of the phase jump point in Figure 8a. From the elevation change curves of the marked line segments in Area 1 and Area 2 shown in Figure 10a,b, it can be seen that the jump between adjacent pixels in the MAPG-inverted DEM reaches 20 m in some areas, which does not correspond to the actual situation. On the other hand, the MAPV-inverted DEM shows better results without significant elevation jumps.

Table 2 evaluates the continuity of Area 1 and Area 2. The criterion used is the accuracy $\alpha$, where discrepancies greater than the set threshold compared to the Alos DEM are deemed as discontinuities. It is evident that compared to the MAPG-inverted DEM, the MAPV-inverted DEM exhibits better continuity.

**Table 2.** Differential comparison of DEM continuity in inversion methods.

| ID | $\alpha_{15}$ | |
|---|---|---|
| | **MAPG** | **MAPV** |
| Area 1 | 87.48% | 99.06% |
| Area 2 | 89.13% | 99.14% |

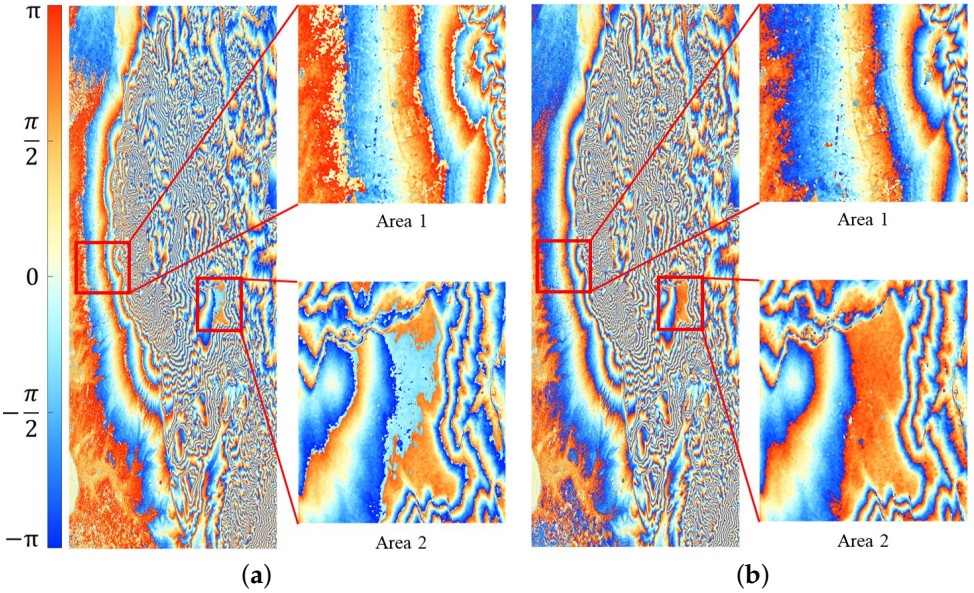

**Figure 8.** Ground phase estimated by (**a**) MAPG. (**b**) MAPV.

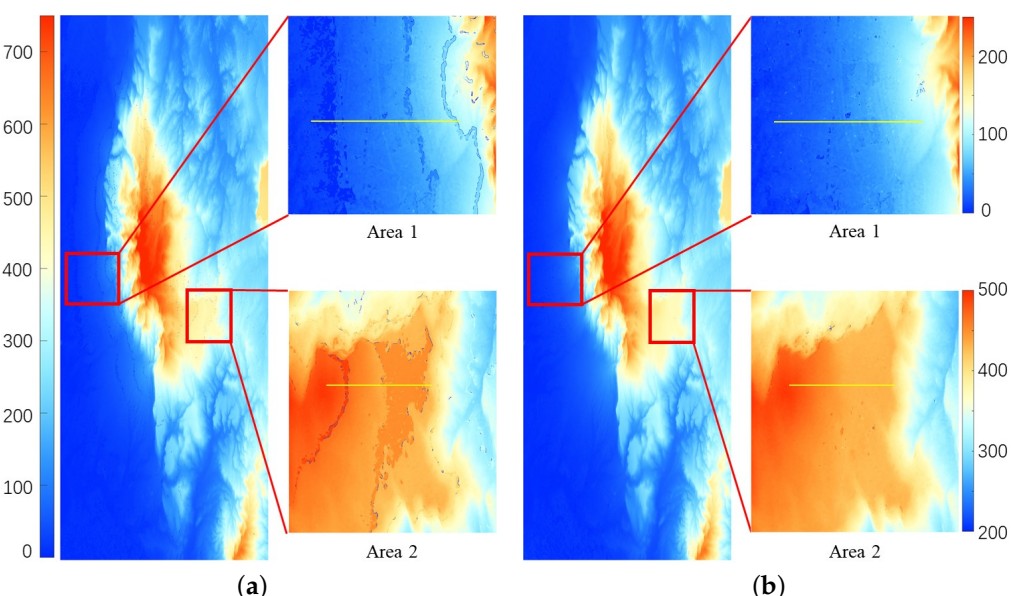

**Figure 9.** (**a**) MAPG-inverted DEM. (**b**) MAPV-inverted DEM.

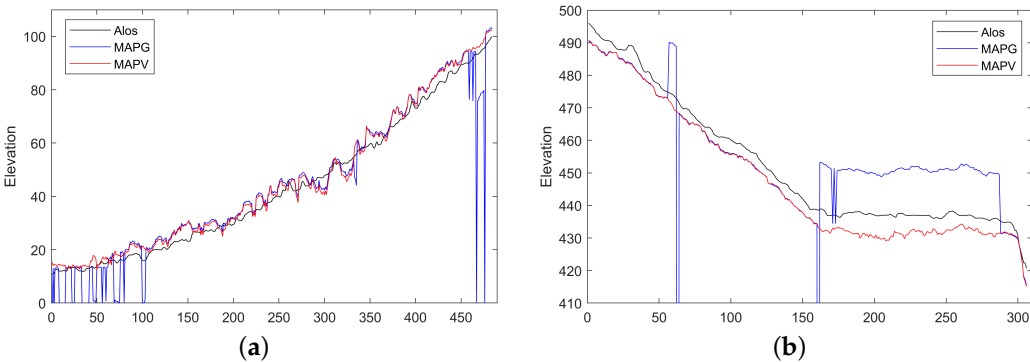

**Figure 10.** (**a**) Section elevation change curve of marked line segment in Area 1. (**b**) Section elevation change curve of marked line segment in Area 2.

*5.4. Performance Assessment of the Proposed Method*

In this section, IceSat-2 data are used to confirm the effectiveness of MAPV in generating DEM for forested areas.

Figure 11 shows the distribution of IceSat-2 data in the test forest area, where the color of the dots indicates the magnitude of the difference between the calculated elevation at that point and the LiDAR data. Figure 11a,b show the difference between the Alos and SRTM DEM with the LiDAR DEM, respectively. With reference to the colorbar on the right, it can be seen that the Alos and SRTM DEM are generally higher than the LiDAR DEM. Figure 11c shows the difference between the TSI-inverted DEM and the LiDAR DEM. It is worth noting that, as mentioned above, the assumptions made by TSI in solving the double-candidate effect cannot satisfy all conditions, and errors will occur in some low-coherence regions, resulting in elevation jumps in these regions when inverting the DEM. Figure 11d shows the difference between the MAPV-inverted DEM and the LiDAR DEM, and it can be seen that the DEM obtained by this method is closer to the LiDAR DEM.

Table 3 shows the mean error (ME) and the root mean square error (RMSE) of the different source DEM compared to the LiDAR DEM. Compared to the DEM of Alos and SRTM, the RMSE of the MAPV-inverted DEM is improved by 23.1% and 24.1%, respectively. It should be noted that due to the elevation jumps in the TSI-inverted DEM, the elevation in some areas may differ significantly from the LiDAR DEM, resulting in a large RMSE.

**Table 3.** Assessment of the elevation accuracy of DEM.

| DEM | ME (*m*) | RMSE (*m*) |
| --- | --- | --- |
| TSI | −13.2407 | 73.7140 |
| ALOS | 2.6370 | 7.7956 |
| SRTM | 2.1311 | 7.8972 |
| MAPV | 0.2111 | 5.9944 |

Figure 12 shows a histogram of the distribution of elevation differences between different source DEM. Obviously, compared to the DEM of Alos, SRTM, and TSI, the error center of MAPV-inverted DEM is closer to 0.

The method proposed in this paper validates the feasibility of using spaceborne L-band PolInSAR for retrieving forest understory DEM. The rapid development of spaceborne PolInSAR technology has expanded the coverage and increased the update speed of PolInSAR data, enabling the generation of large-scale, accurate DEM. High-precision, rapidly updated DEM plays a crucial role in fields such as environmental monitoring and disaster monitoring, providing precise data support for assessing forest coverage and terrain changes, among other issues. However, solving for the ground phase requires high-quality data, with the interferometric pair needing suitable temporal and spatial baselines, as well as good coherence. Low-quality data can result in inaccurate phase solutions, leading to errors in the inverted

DEM. Furthermore, future research needs to further explore the applicability of this method under different climatic conditions or forest types and address potential challenges, such as considering the impact of varying climatic conditions on data quality and investigating interference effects and L-band penetration depth in different types of forests.

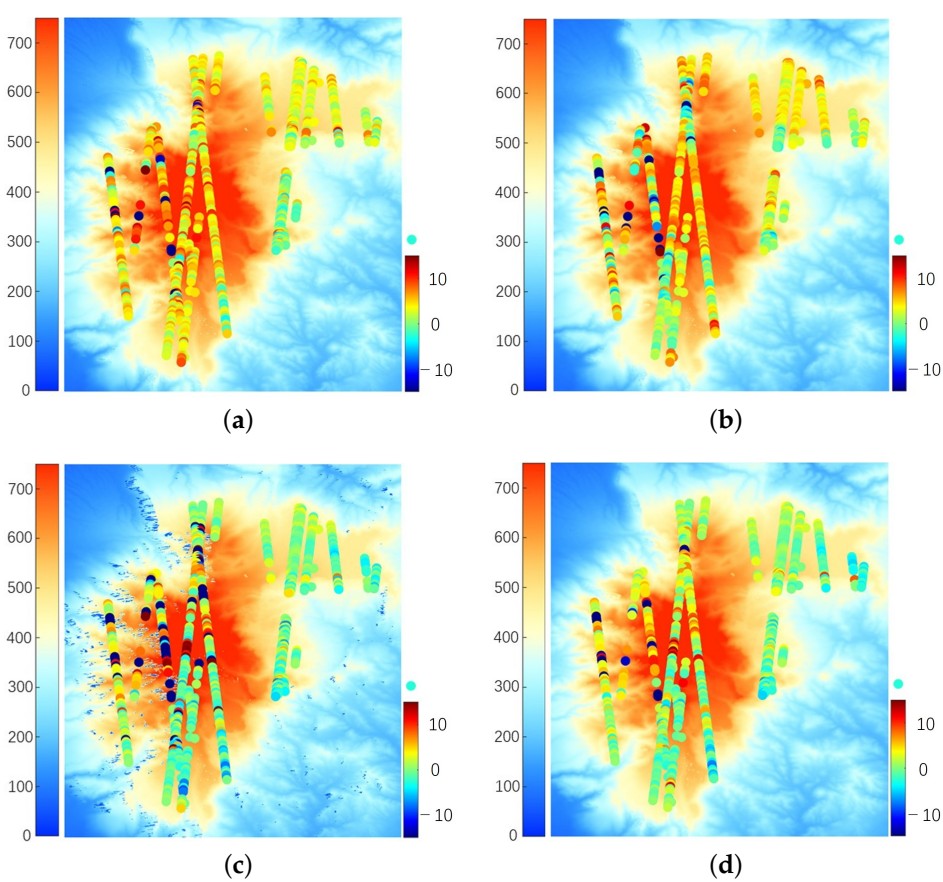

**Figure 11.** The geocoded DEM maps and the difference between the LiDAR DEM and the DEM from different sources. (**a**) Alos-30m. (**b**) SRTM-30m (**c**) TSI. (**d**) MAPV.

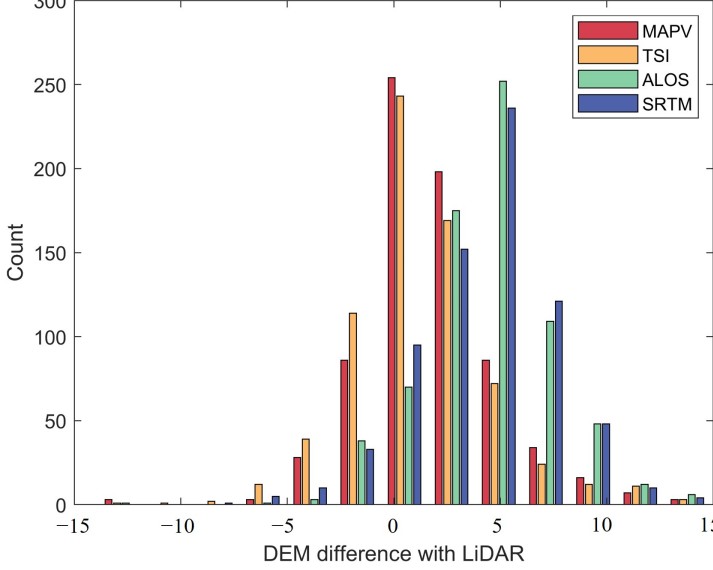

**Figure 12.** Histogram of the difference between the DEM obtained through different methods and the LiDAR DEM.

## 6. Conclusions

This paper proposes a MAPV inversion algorithm for RVoG model, and generates high-precision radar DEM data in forested area. The method is based on maximum a posteriori estimation and models the ground phase as a Von Mises distribution, with its mean derived from the topographic phase computed from external DEM. This method effectively overcomes the double-candidate effect, addresses the phase jump issue caused by modeling ground phases as a Gaussian distribution, and ensures the continuity and accuracy of ground phase solutions. Additionally, this paper derives, analyzes, and simulates the CRLB for MAPV. The results indicate that the introduction of prior information significantly reduces the CRLB of this method. In response to the characteristics of the MAPV objective function, this paper proposes a FSO method to enhance the efficiency of ground phase determination. FSO obtains two candidate phases separately based on the gradient descent algorithm and selects the one with the higher function value as the ground phase, as the gradient descent algorithm requires setting a global learning rate and there exist significant differences in the objective functions among different pixels. This paper improves the traditional RMSprop algorithm by reducing its dependency on the global learning rate, enabling it to handle various objective functions.

In this paper, the L-band spaceborne SAOCOM data are used to estimate the PolInSAR understory DEM to evaluate and verify the effectiveness of the method. The results demonstrate that:

1. Compared to the traditional exhaustive search method, FSO significantly improves computational efficiency without compromising accuracy.
2. The MAPV method effectively addresses the issue of elevation jumps in DEM caused by the discontinuity in ground phase solutions by MAPG.
3. Using IceSat-2 data as a benchmark, the DEM of the test forest area is compared with the DEMs of Alos, SRTM, and TSI. The results show that MAPV has better estimation performance, with improvements in both mean error (ME) and root mean square error (RMSE).

**Author Contributions:** Conceptualization, X.L. (Xiaoshuai Li); methodology, X.L. (Xiaoshuai Li); software, X.L. (Xiaoshuai Li); validation, X.L. (Xiaoshuai Li); formal analysis, X.L. (Xiaoshuai Li); investigation, X.L. (Xiaoshuai Li), X.L. (Xiaolei Lv) and Z.H.; resources, X.L. (Xiaoshuai Li), X.L. (Xiaolei Lv) and Z.H.; data curation, X.L. (Xiaoshuai Li); writing—original draft preparation, X.L. (Xiaoshuai Li); writing—review and editing, X.L. (Xiaoshuai Li), X.L. (Xiaolei Lv) and Z.H.; visualization, X.L. (Xiaoshuai Li); supervision, X.L. (Xiaolei Lv) and Z.H.; project administration, X.L. (Xiaolei Lv); funding acquisition, X.L. (Xiaolei Lv). All authors have read and agreed to the published version of the manuscript.

**Funding:** This research was funded by the LuTan-1 L-Band Spaceborne Bistatic SAR data processing program, grant number E0H2080702.

**Data Availability Statement:** In this study, the remote sensing data were obtained from various sources to support our analyses. We accessed Satélite Argentino de Observación COn Microondas (SAOCOM) at http://saocom.asi.it:8081/#/home (accessed on 5 March 2024), Ice, Cloud and land Elevation Satellite-2 (ICESat-2) at https://nsidc.org/data/icesat-2 (accessed on 5 March 2024), Advanced Land Observing Satellite (ALOS) at https://www.eorc.jaxa.jp/ALOS/en/aw3d30/data/index.htm (accessed on 5 March 2024), Shuttle Radar Topography Mission 30 m (SRTM30) at https://earthexplorer.usgs.gov/ (accessed on 5 March 2024). These diverse data sources played a crucial role in our research and provided a comprehensive foundation for our remote sensing investigations.

**Conflicts of Interest:** The authors declare no conflicts of interest.

## Abbreviations

The following abbreviations are used in this manuscript:

| | |
|---|---|
| DEM | Digital Elevation Model |
| PolInSAR | Polarimetric Interferometric Synthetic Aperture Radar |
| Lidar | Light Detection and Ranging |
| RVoG | Random Volume over Ground |
| TSI | Three-Stage Inversion |
| MAP | Maximum a Posteriori |
| MAPG | Maximum a Posteriori with Gaussian distribution as prior |
| MAPV | Maximum a Posteriori with Von Mises distribution as prior |
| CRLB | Cramer–Rao Lower Bound |
| FSO | Four-Step Optimization |
| ME | Mean Error |
| RMSE | Root Mean Square Error |

## Appendix A. The Fisher Information of MAPV

It can be inferred from (14) that the Fisher information consists of two parts, with the first part provided by the RVoG model,

$$
\begin{aligned}
I_{F_{RVoG}} &= -E\left[\frac{\partial^2 \log P(\hat{\boldsymbol{R}}; \boldsymbol{R}_n, N)}{\partial \phi^2}\right] \\
&= -E\left[\frac{\partial^2(\log C(\hat{\boldsymbol{R}}) - N\log|\boldsymbol{R}_n| - Ntr(\boldsymbol{R}_n^{-1}\hat{\boldsymbol{R}}))}{\partial \phi^2}\right] \\
&= -E\left[-\frac{N}{|\boldsymbol{R}_n|}\frac{\partial|\boldsymbol{R}_n|}{\partial \phi} - N\frac{\partial tr(\boldsymbol{R}_n^{-1}\hat{\boldsymbol{R}})}{\partial \phi}\right] \\
&= -E\left[-Ntr(\boldsymbol{R}_n^{-1}\frac{\partial \boldsymbol{R}_n}{\partial \phi}) - Ntr(-\boldsymbol{R}_n^{-1}\frac{\partial \boldsymbol{R}_n}{\partial \phi}R^{-1}\hat{\boldsymbol{R}})\right] \\
&= -E\left[-Ntr\left(\frac{\partial \boldsymbol{R}_n^{-1}}{\partial \phi}\frac{\partial \boldsymbol{R}_n}{\partial \phi} + \boldsymbol{R}_n^{-1}\frac{\partial^2 \boldsymbol{R}_n}{\partial \phi^2}\right)\right. \\
&\quad \left. +Ntr\left(\left(2\frac{\partial \boldsymbol{R}_n^{-1}}{\partial \phi}\frac{\partial \boldsymbol{R}_n}{\partial \phi}\boldsymbol{R}_n^{-1} + \boldsymbol{R}_n^{-1}\frac{\partial^2 R}{\partial \phi^2}\boldsymbol{R}_n^{-1}\right)\hat{\boldsymbol{R}}\right)\right] \\
&= -E\left[-Ntr\left(-\boldsymbol{R}_n^{-1}\frac{\partial \boldsymbol{R}_n}{\partial \phi}\boldsymbol{R}_n^{-1}\frac{\partial \boldsymbol{R}_n}{\partial \phi} + \boldsymbol{R}_n^{-1}\frac{\partial^2 \boldsymbol{R}_n}{\partial \phi^2}\right)\right. \\
&\quad \left. +Ntr\left(\left(-2\boldsymbol{R}_n^{-1}\frac{\partial \boldsymbol{R}_n}{\partial \phi}\boldsymbol{R}_n^{-1}\frac{\partial \boldsymbol{R}_n}{\partial \phi}\boldsymbol{R}_n^{-1} + \boldsymbol{R}_n^{-1}\frac{\partial^2 \boldsymbol{R}_n}{\partial \phi^2}\boldsymbol{R}_n^{-1}\right)\hat{\boldsymbol{R}}\right)\right] \\
&= -Ntr\left(\boldsymbol{R}_n^{-1}\frac{\partial \boldsymbol{R}_n}{\partial \phi}\boldsymbol{R}_n^{-1}\frac{\partial \boldsymbol{R}_n}{\partial \phi}\right) + Ntr\left(\boldsymbol{R}_n^{-1}\frac{\partial^2 \boldsymbol{R}_n}{\partial \phi^2}\right) \\
&\quad + 2Ntr\left(\boldsymbol{R}_n^{-1}\frac{\partial \boldsymbol{R}_n}{\partial \phi}\boldsymbol{R}_n^{-1}\frac{\partial \boldsymbol{R}_n}{\partial \phi}\right) - Ntr\left(\boldsymbol{R}_n^{-1}\frac{\partial^2 \boldsymbol{R}_n}{\partial \phi^2}\right) \\
&= Ntr\left(\boldsymbol{R}_n^{-1}\frac{\partial \boldsymbol{R}_n}{\partial \phi}\boldsymbol{R}_n^{-1}\frac{\partial \boldsymbol{R}_n}{\partial \phi}\right)
\end{aligned}
\tag{A1}
$$

and the second part being the information incorporated from the prior,

$$
\begin{aligned}
I_{\mathrm{F_{VM}}} &= -E\left[\frac{\partial^2 \log p(\phi; \phi_{\mathrm{topo}}, \kappa)}{\partial \phi^2}\right] \\
&= -E\left[\frac{\partial^2 (-\log 2\pi I_0(\kappa) + \kappa \cos(\phi - \phi_0))}{\partial \phi^2}\right] \\
&= -E[-\kappa \cos(\phi - \phi_0)] \\
&= \int_{-\pi}^{\pi} \frac{1}{2\pi I_0(\kappa)} \kappa \cos(\phi - \phi_0) e^{\kappa \cos(\phi - \phi_0)} d\phi \\
&= \frac{\kappa I_1(\kappa)}{I_0(\kappa)}.
\end{aligned}
\tag{A2}
$$

Thus, the Fisher information for the ground phase $\phi$ can be expressed as

$$
\begin{aligned}
I_{\mathrm{F}} &= I_{\mathrm{F_{RVoG}}} + I_{\mathrm{F_{VM}}} \\
&= Ntr\left(\boldsymbol{R}_n^{-1}\frac{\partial \boldsymbol{R}_n}{\partial \phi}\boldsymbol{R}_n^{-1}\frac{\partial \boldsymbol{R}_n}{\partial \phi}\right) + \frac{\kappa I_1(\kappa)}{I_0(\kappa)}.
\end{aligned}
\tag{A3}
$$

**Appendix B. Gradient of the MAPV Objective Function**

From (17), we have

$$
\frac{\partial f}{\partial \phi} = \frac{3\sin\theta}{1 - \cos\theta}\cdot\frac{\partial \theta}{\partial \phi} - \frac{1}{|A_{\theta+\phi}|}\cdot\frac{\partial |A_{\theta+\phi}|}{\partial(\theta+\phi)}\cdot\frac{\partial(\theta+\phi)}{\partial \phi} - \frac{1}{|A_\phi|}\cdot\frac{\partial |A_\phi|}{\partial \phi} - k\sin(\phi - \phi_0)/N
\tag{A4}
$$

where [46]

$$
\frac{\partial |A_\alpha|}{\partial \alpha} = |A_\alpha| tr\left(A_\alpha^{-1}\frac{\partial A_\alpha}{\partial \alpha}\right)
\tag{A5}
$$

and from (16),

$$
\frac{\partial A_\alpha}{\partial \alpha} = \frac{j}{2}\left(e^{-j\alpha}\hat{\boldsymbol{\Omega}} - e^{j\alpha}\hat{\boldsymbol{\Omega}}^H\right).
\tag{A6}
$$

From (19),

$$
\begin{aligned}
\frac{\partial \theta}{\partial \phi} &= \frac{\partial\left(2\arctan\left[-3\left(\frac{|A_\phi|'}{A_\phi} + \frac{k}{N}\sin(\phi - \phi_{\mathrm{topo}})\right)^{-1}\right]\right)}{\partial \phi} \\
&= \frac{2}{1 + 9\left(\frac{|A_\phi|'}{|A_\phi|} + \frac{k}{N}\sin(\phi - \phi_{\mathrm{topo}})\right)^{-2}}\cdot\frac{\partial\left(-3\left(\frac{|A_\phi|'}{|A_\phi|} + \frac{k}{N}\sin(\phi - \phi_{\mathrm{topo}})\right)^{-1}\right)}{\partial \phi} \\
&= \frac{2\left(\frac{|A_\phi|'}{|A_\phi|} + \frac{k}{N}\sin(\phi - \phi_{\mathrm{topo}})\right)^2}{\left(\frac{|A_\phi|'}{|A_\phi|} + \frac{k}{N}\sin(\phi - \phi_{\mathrm{topo}})\right)^2 + 9}\cdot\frac{3\left(\frac{|A_\phi|''|A_\phi| - |A_\phi|'^2}{|A_\phi|^2} + \frac{k}{N}\cos(\phi - \phi_{\mathrm{topo}})\right)}{\left(\frac{|A_\phi|'}{|A_\phi|} + \frac{k}{N}\sin(\phi - \phi_{\mathrm{topo}})\right)^2} \\
&= \frac{6\left(\frac{|A_\phi|''|A_\phi| - |A_\phi|'^2}{|A_\phi|^2} + \frac{k}{N}\cos(\phi - \phi_{\mathrm{topo}})\right)}{\left(\frac{|A_\phi|'}{|A_\phi|} + \frac{k}{N}\sin(\phi - \phi_{\mathrm{topo}})\right)^2 + 9}
\end{aligned}
\tag{A7}
$$

where $|A_\phi|''$ represents the second derivative of $|A_\phi|$ with respect to $\phi$,

$$
\begin{aligned}
|A_\phi|'' &= |A_\phi|' tr\left(A_\phi^{-1}\frac{\partial A_\phi}{\partial \phi}\right) + |A_\phi| tr\left(\frac{\partial A_\phi^{-1}}{\partial \phi}\frac{\partial A_\phi}{\partial \phi} + A_\phi^{-1}\frac{\partial^2 A_\phi}{\partial \phi^2}\right)\\
&= |A_\phi| tr^2\left(A_\phi^{-1}\frac{\partial A_\phi}{\partial \phi}\right) + |A_\phi| tr\left(-A_\phi^{-1}\frac{\partial A_\phi}{\partial \phi}A_\phi^{-1}\frac{\partial A_\phi}{\partial \phi} + A_\phi^{-1}\frac{\partial^2 A_\phi}{\partial \phi^2}\right)
\end{aligned}
\tag{A8}
$$

and

$$
\frac{\partial^2 A_\phi}{\partial \phi^2} = \frac{1}{2}\left(e^{-j\phi}\hat{\Omega} + e^{j\phi}\hat{\Omega}^H\right).
\tag{A9}
$$

Substituting (A8) into (A7), we can obtain that

$$
\frac{\partial \theta}{\partial \phi} = \frac{6\left(tr\left(-A_\phi^{-1}\frac{\partial A_\phi}{\partial \phi}A_\phi^{-1}\frac{\partial A_\phi}{\partial \phi} + A_\phi^{-1}\frac{\partial^2 A_\phi}{\partial \phi^2}\right) + \frac{k}{N}\cos(\phi - \phi_{\text{topo}})\right)}{\left(tr\left(A_\phi^{-1}\frac{\partial A_\phi}{\partial \phi}\right) + \frac{k}{N}\sin(\phi - \phi_{\text{topo}})\right)^2 + 9}.
\tag{A10}
$$

Therefore,

$$
\frac{\partial f}{\partial \phi} = \frac{3\sin\theta}{1 - \cos\theta}\frac{\partial \theta}{\partial \phi} - tr\left(A_{\theta+\phi}^{-1}\frac{\partial A_{\theta+\phi}}{\partial(\theta+\phi)}\right)\frac{\partial(\theta+\phi)}{\partial \phi} - tr\left(A_\phi^{-1}\frac{\partial A_\phi}{\partial \phi}\right) - \frac{k}{N}\sin(\phi - \phi_0)
\tag{A11}
$$

the gradient of the MAPV objective function can be obtained by substituting (A6), (A9), and (A10) into (A11).

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
