# Peer review of "Underlying Topography Estimation over Forest Using Maximum a Posteriori Inversion with Spaceborne Polarimetric SAR Interferometry"

_remotesensing, doi:10.3390/rs16060948_

Round 1

Reviewer 1 Report

Comments and Suggestions for Authors

Dear Editor,

Thank you for affording me the chance to review the manuscript titled "Underlying Topography Estimation over Forest Using Maximum a Posteriori Inversion with Spaceborne Polarimetric SAR Interferometry" by Xiaoshuai Li and his/her colleagues, which has been submitted to the journal "Remote Sensing"

This paper presents a new method for extracting digital elevation models (DEMs) from Polarimetric Interferometric Synthetic Aperture Radar (PolInSAR) data, specifically in forested areas. It introduces an innovative approach using the Von Mises distribution for ground phase modeling and a Four-Step Optimization method for solving the ground phase efficiently. The study validates its methodology using spaceborne L-band SAOCOM data, demonstrating significant improvements in efficiency and accuracy over traditional methods. The research contributes to the field of remote sensing and geospatial analysis, particularly in enhancing the accuracy of DEMs in challenging environments like forests.

In general, the paper is well-structured with clear results and rigorous logic. Here are some suggestions that may further enhance the quality of the paper:

1. A potential area for improvement in the introduction could be the inclusion of a more detailed discussion on the practical applications and significance of the improved DEM estimation in forested areas. While the technical challenges and solutions are well-described, expanding on how these improvements could benefit specific fields like environmental monitoring, forestry management, or land use planning would provide a stronger context for the study's relevance. This addition would help readers better understand the broader implications and potential impact of the research.

2. I understand that, in the "Maximum a Posteriori Estimation of the Ground Phase" section, the paper provides a detailed and technical explanation of the MAP estimation process for the ground phase in the context of the RVoG model. It introduces the concept of Von Mises distribution as a prior in the estimation process to address the limitations of the Gaussian distribution, especially near phase jump points.

One area for improvement could be in the explanation and justification of the choice of the Von Mises distribution over other potential distributions. While the paper mentions its advantages over the Gaussian distribution, a more detailed comparison or discussion about why the Von Mises distribution is the most suitable choice for this application would be beneficial. This could include a more thorough analysis of its properties, how it specifically addresses the challenges posed by phase wrapping, and why other distributions might not be as effective. Such an explanation would not only bolster the methodological choices made but also provide the reader with a deeper understanding of the underlying statistical considerations.

3. My suggestion for the "Results and Discussion" section of the paper is to enhance the discussion on the implications and real-world applications of the findings. For example, while the paper effectively presents the quantitative improvements in DEM accuracy using the new method, it could further discuss how these improvements can be practically applied in fields like environmental monitoring, where accurate DEMs are crucial for assessing changes in forest cover or topography. Additionally, discussing potential challenges in implementing this method in various real-world scenarios, such as in different climatic conditions or forest types, would add depth to the paper. These discussions would provide readers with a more comprehensive understanding of the practical significance and limitations of the research.

Reviewer 2 Report

Comments and Suggestions for Authors

A remarkable paper represents the use of PolInSAR technology for DEM extraction in forested areas. The applied methodological approach improves the quality of the outputs obtained from the "Polarimetric SAR Interferometry" data. The structure of the paper is well laid out. I appreciate the illustrative, graphical and detailed description of the principles and partial methodological procedures, including the capture of the flowchart process and the refinement of more complex equations in the Appendix. The citation apparatus has an adequate number of titles reflecting the methodological and thematic focus of the paper.

There is a slight reservation about Figure 4. It assumed that graph b) represents a detail of the cutout of graph a). However, this needs to be apparent from the description of the graph in the text, or for clarity, it would be useful to define it at least in the caption below the figure.

The description of equation 21 on line 213 defines a variable that is missing from the notation (σ2pq).

Except for a minor comment regarding Fig. 4 and equation 21, I have no other reservations about the article.

Reviewer 3 Report

Comments and Suggestions for Authors

Dear Authors,

You presented an overall well-organized manuscript, synthetic and clear in the steps toward the objective. I have a few comments:

One suggestion would be to move the materials before the long methods chapter. Moreover, I would add at the very beginning, for further understanding, a workflow map of the used methodology (either Figure 3 seems ok). 

I would put, if it is possible, the list of abbreviations at the beginning to help the reader.

- a missing space in lines 20 and 47;

- line 41: I would specify also here what RVoG stands for
